# Inhibition of Liver Tumor Cell Metastasis by Partially Acetylated Chitosan Oligosaccharide on A Tumor-Vessel Microsystem

**DOI:** 10.3390/md17070415

**Published:** 2019-07-13

**Authors:** Bolin Jing, Gong Cheng, Jianjun Li, Zhuo A. Wang, Yuguang Du

**Affiliations:** 1State Key Laboratory of Biochemical Engineering, Institute of Process Engineering, Chinese Academy of Sciences, Beijing 100190, China; 2University of Chinese Academy of Sciences, Beijing 100049, China

**Keywords:** partially acetylated chitooligosaccharides, liver tumor cell metastasis, dynamic tumor-vessel microsystem

## Abstract

Chitooligosaccharides (COS), the only cationic oligosaccharide in nature, have been demonstrated to have anti-tumor activity. However, the inhibitory effects of COS on different stages of tumor metastasis are still unknown, and it is not clear what stage(s) of tumor metastasis COS targeted. To study the inhibitory effects of a new partially acetylated chitooligosaccharide (paCOS) with fraction of acetylation (F_A_) 0.46 on each phase of liver cancer cell metastasis, a dynamic tumor-vessel microsystem undergoing physiological flow was leveraged. paCOS (F_A_ = 0.46) significantly inhibited proliferation of HepG2 cells through vascular absorption on the chip, and inhibited migration of HepG2 cells by inhibiting the formation of pseudopod in liver tumor cells. It was also found that paCOS at 10 μg/mL had a stronger inhibitory effect on liver tumor cells invading blood vessels than that of paCOS at 100 μg/mL, and paCOS at 100 μg/mL, which had a significant destructive effect on tumor vascular growth and barrier function. Moreover, paCOS reduced the number of liver tumor cells adhering onto the surface of HUVECs layer after 3 h of treatment. Therefore, the results revealed that paCOS had considerable potential as drugs for anti-tumor metastasis.

## 1. Introduction

Hepatocellular carcinoma (HCC) is one of the leading causes of cancer-related deaths worldwide [1,2]. More than 700,000 new cancer cases are diagnosed each year throughout the world and also more than 600,000 deaths are attributed to HCC each year [3]. It has been proven that metastasis is responsible for as much as 90% of cancer-associated mortality [4]. It is well known that metastatic dissemination of cancer cells is a highly complex and multi-step biological process starting with invasion of cancer cells through the extracellular matrix (ECM) of the stroma toward the blood and lymph vessels [5]. Evaluation of the inhibitory effects of drugs on different stages of tumor metastasis (proliferation, migration, intravasation, adhesion) is important for developing more accurate, target-specific antitumor drugs. However, it has been challenging to do this because it is not possible to independently simulate each stages of tumor metastasis in current tumor metastasis models. For instance, it is difficult for animal models to identify each stage of tumor metastasis in a complex physiological environment. Traditional static cell culture cannot mimic the process of tumor cells invading into blood vessels because it is not easy to achieve three-dimensional co-culture of tumor cells and stromal cells, including endothelial cells, fibroblasts, and so on [6,7]. 

In the past decade, microfluidic technology with evident advantages, such as a small sample volume, high sensitivity, fast processing speed, high portability and low cost, has become an increasingly promising tool for basic and applied research on cancer [8,9]. The use of microfluidic chips could greatly increase the potential for precise control of parameters and accurately mimic the tumor microenvironment for studying tumor cell metastasis and screening anticancer drugs. For example, Bersini et al. developed a 3D microfluidic model to analyze the specificity of human breast cancer metastasis to bone [10]. Mi and co-workers presented a novel microfluidic co-culture system and established mild, moderate and severe cancer models by using HMEpiC and MDA-MB–231 cells to study cancer cell migration and screen anti-cancer drugs [11]. Chen et al. developed a microfluidic platform to mimic the physiological microenvironment of solid tumors with multicellular tumor spheroids for screening anticancer drugs [12]. We have developed a dynamic tumor-vessel microfluidic device that enabled co-culture tumor cells and endothelial cells at physiological relevant luminal flow rates, which could reproduce the different phases of cancer metastasis individually [13].

Chitosan oligosaccharides (COS) are oligosaccharides with a degree of polymerization (DP) of <20 and an average molecular weight (MW) of <3.9 kDa, which are produced enzymatically or chemically from chitosan [14]. Chitosan is a linear polysaccharide consisting of β-1,4-d-glucosamine (GlcN, D) and β-1,4-*N*-acetyl-d-glucosamine (GlcNAc, A) [15]. The main source of chitosan is from the deacetylation of chitin, which is the major component of exoskeletons of shellfish and cell walls of fungi [16]. COS have various physiological activities such as antibacteria, antivirus, antitumor, anti-oxidation, immune regulation, and so on [17,18,19,20,21]. The anti-tumor effects of COS have already been known for several decades. In 1986, Suzuki et al. reported that both fully acetylated and fully deacetylated chitohexaose possessed anti-tumor activity [22]. Since then, different aspects of the anti-tumor activity of COS were studied, such as the effects of MW or DP [23,24] and fraction of acetylation (F_A_) [25]. These data indicated that MW or DP and F_A_ of COS were important factors for suppressing growth of cancer cells. Fox example, partially acetylated COS (paCOS) with F_A_ 0.3 gave the highest anti-angiogenic activity [26]. Furthermore, it was reported that F_A_ is more important than DP for the antitumor effects of paCOS [26]. However, due to the limitations of animal models and traditional static in vitro models, it remains unknown which phase(s) of tumor metastasis COS or paCOS targeted.

In this study, paCOS with F_A_ 0.46 were prepared from partially acetylated chitosan with recombinant endo-chitosanase (CSN) from *Aspergillus fumigatus* overexpressed in *Pichia pastoris* [27]. The tumor-vessel microsystem was leveraged to investigate the anti-metastatic effects of paCOS (F_A_ = 0.46) on each phase of liver tumor cell metastasis (proliferation, migration, intravasation and adherence). 

## 2. Results 

### 2.1. Preparation and Characterization of paCOS 

Partially acetylated chitosan was prepared by following the published procedure, and its F_A_ was determined to be 0.46 by ^1^H NMR. paCOS was prepared by enzymatic hydrolysis of partially acetylated chitosan using endo-chitosanase (CSN) from *Aspergillus fumigatus* produced in *Pichia pastoris,* and F_A_ of paCOS was supposed to be same as that of the chitosan substrate. The result of MALDI-TOF-MS revealed that structures of hydrolysis products mainly included paCOS with DPs in the range of 2 to 9 and including one fully deacetylated component (D_3_, GlcN_3_) (Figure 1). For paCOS with same DP, their structures and components were different. For example, if the exact sequence of each paCOS was not considered, there might be at least three types of tetrasaccharides for paCOS with DP = 4: A1D3, A2D2, and A3D1, whereas there may be at least four types of pentasaccharide for paCOS with DP = 5: A1D4, A2D3, A3D2 and A4D1. If the exact sequence of each paCOS was taken into account, there would be more isomers. For instance, there might be five isomers for tetrasaccharide A1D4 only: ADDDD, DADDD, DDADD, DDDAD and DDDDA. The exact sequence of each paCOS needs further characterization, which is not the main focus of the current study. 

### 2.2. Inhibitory Effects of paCOS on Liver Tumor Cell Proliferation

To determine whether paCOS with F_A_ 0.46 could suppress proliferation of liver tumor cells, the dynamic tumor-vessel microfluidic model for proliferation of tumor cells was employed [13]. The inhibitory effects of paCOS at different concentrations on the proliferation of HepG2 cells were shown in Figure 2A. The proliferation rate of HepG2 cells in the negative control group (incubated in culture medium without paCOS) was regarded as 100%, inhibitory effects of paCOS on HepG2 cells were positively related to paCOS concentrations. The highest inhibitory effect of paCOS on the proliferation rate of HepG2 cells was 91.6 ± 1.7% at 100 µg/mL, which was slightly higher than that of 5-Fu as the positive control (89.9 ± 5.0%) at the same concentration (Figure 2B). Besides, we have studied the inhibitory effects of paCOS at different concentration on the proliferation of two other liver cancer cells (SMMC-7721 cells and MHCC97-L cells) using the dynamic tumor-vessel microfluidic model. As shown in Appendix A, the proliferation rates of SMMC-7721 cells and MHCC97-L cells treated with 100 µg/mL paCOS were 45.1 ± 7.3% and 55.2 ± 2.6%, which were much lower than that of HepG2 cells (91.6 ± 1.7%).

The cytotoxicity of paCOS with F_A_ 0.46 on liver cancer cells was also evaluated by determining apoptosis of HepG2 cell. As shown in Figure 2C, the number of apoptotic cells per square millimeter increased with the increase of paCOS concentrations. paCOS showed apoptotic inducing activity on liver cancer cells even at 1 µg/mL. Besides, the highest apoptosis number of HepG2 cells treated by paCOS at 100 µg/mL was 167 ± 15 cells/mm^2^, which was close to that of 5-Fu (204 ± 28 cells/mm^2^) at the same concentration (Figure 2D). 

### 2.3. Inhibitory Effects of paCOS on Liver Tumor Cell Migration

In order to study the inhibitory effects of paCOS with F_A_ 0.46 on the migration ability of liver tumor cells, tumor cell clusters packaged by extracellular matrix were seeded on the endothelial monolayer within the micro-device. By tracking HepG2 cells labeled by green cell-tracker, it was found that the migration ability of liver tumor cells was significantly inhibited by paCOS (Figure 3A). The migration rate of HepG2 cells in the negative control group (incubated in culture medium without paCOS) was regarded as 100%, and the migration rate of liver tumor cells treated by 100 µg/mL paCOs was 95.1 ± 1.8%, which was higher than that of 5-Fu (88.2 ± 3.5%) at the same concentration (Figure 3B).

The inhibitory effects of different paCOS concentrations on the migration ability of liver cancer cells were also evaluated by comparing the distance of tumor cells disseminated to a scratched area on the 24-well plate with or without paCOS treatment. Statistical analysis showed that, the inhibitory effects of paCOS on migration of HepG2 cells were positively related to the paCOS concentrations (Figure 3C). The average migration distance of HepG-2 cells was 170 ± 10 μm without paCOS treatment, and the migration distance of HepG-2 cells was only 20 ± 5 μm when the HepG-2 cells were treated with 100 µg/mL of paCOS (Figure 3D). 

### 2.4. Inhibitory Effects of paCOS on Liver Tumor Cell Intravasation and Invasion

To examine the inhibitory effect of paCOS with F_A_ 0.46 on intravasation of liver cancer cells, the number of HepG-2 cells invading into the vascular lumen was counted after 24 h of co-cultivation (Figure 4A). Compared with the negative control group (no inhibitor present), the intravasation rate of HepG2 cells treated with 10 µg/mL paCOS was 83 ± 4%, which was higher than that of paCOS at 100 µg/mL (78 ± 4%) (Figure 4B). 

To investigate the effects of paCOS on the barrier function of vascular endothelial layer, the HE staining and measurement of the *P_app_* value of FITC-dextran (10 kDa) across endothelial cells were done after 24 h of intravasation (Figure 4C,D). It was interesting to find that the density of vascular endothelial was the highest and the *P_app_* value of endothelium was the lowest (1.65 ± 0.08 × 10^−5^ cm/s) at 10 µg/mL of paCOS. In addition, as shown in Appendix A, it was found that paCOS at > 100 μg/mL had significant inhibitory effect on the viability of vascular endothelial cells (< 89.2 ± 2.6%) and destructive effect on permeability of vascular endothelial cells (> 2.27 ± 1.86 × 10^−5^ cm/s) in the dynamic microfluidic device.

The inhibitory effects of paCOS on the invasion of liver cancer cells were evaluated by seeding HepG-2 cells coated with extracellular matrix into the Transwell (Appendix A). The invasion rate observed in the absence of inhibitors was set to 100% and the inhibitory effects of paCOS on the invasion rate of HepG-2 cells were positively related to the paCOS concentrations (Appendix A). The invasion rate of HepG-2 cells treated with 100 µg/mL paCOS was 55 ± 4%, which was lower than that of 5-Fu at the same concentration (71 ± 3%).

### 2.5. Inhibitory Effects of paCOS on Liver Tumor Cell Adhesion

To determine the inhibitory effects of paCOS with F_A_ 0.46 on adhesion of liver cancer cells onto endothelium, HepG-2 cells labeled by green cell-tracker were injected into the vascular cavity at a defined flow rate (Figure 5A). Results indicated that paCOS could reduce the number of HepG-2 cells adhering onto the endothelial layer effectively (Figure 5B). Interestingly, the diameter of the tumor cells in the negative control group (in the absence of inhibitors) was over 25 μm, which was longer than those treated with paCOS or 5-Fu (15 μm) (Figure 5C). Thereby both paCOS and 5-Fu could reduce the adhesion ability of tumor cells.

## 3. Discussion

Recently, many reports have indicated that COS possess anti-tumor activities, and F_A_ of paCOS is an important factor for suppressing cancer cell growth [26,28]. However, due to the limitations of animal models and traditional two-dimensional in vitro models, most studies focused on prevention and/or intervention of early stages of carcinogenesis and the growth of tumor cells before the occurrence of invasive malignant diseases. It is not clear how COS or paCOS affected each phase of tumor metastasis. Here, paCOS with F_A_ 0.46 was prepared from partially deacetylated chitin with recombinantly overexpressed chitosanase in *P. pastoris*, and the anti-tumor effects of paCOS on each phase of liver cancer cell metastasis were evaluated for the first time by using a tumor-vessel microsystem.

The results of proliferation assay showed that the antitumor growth effects of paCOS with F_A_ 0.46 on the dynamic tumor-vessel microsystem were stronger than those in the static 96-well plates. These results confirmed that paCOS with F_A_ 0.46 could be transported by endothelial cells with fluid flow during the proliferative phase of tumor cells. The structural units of paCOS include d-glucosamine (d-GlcN), which is easily protonated under physiological conditions, and can be absorbed onto tumor cells with the help of electrostatic interaction by changing the permeability of tumor cells [25,29]. The Hoechst apoptosis assay confirmed that the inhibitory effects of paCOS with F_A_ 0.46 on the growth of hepatoma cells might be related to apoptosis. Previous reports suggested that COS might induce apoptosis by up-regulating the expression of Bcl-2 (a potent anti-apoptotic protein) and down-regulating the expression of Caspase-3 (a key enzyme apoptotic cascade in mammals) [30,31]. 

Both dynamic migration assay on the microsystem and wound-healing assay on 24-well pates confirmed that paCOS with F_A_ 0.46 had significant inhibitory effects on the migration ability of liver tumor cells (HepG-2). Morphological analysis showed that tumor cells in the negative control group adopted heterogeneous morphology with round and elongated shapes, and migrated toward the outer regions of seeded cancer cells. Similar to the effects of 5-Fu, paCOS with F_A_ 0.46 could inhibit the extension of the pseudopod and the establishment of new adhesion of the liver tumor cells [32]. 

Intravasation of tumor cells into blood vessels was an important process of tumor metastasis, which was characterized by the migration of tumor cells from the 3D matrix across the basal endothelial surface and the subsequent appearance on the apical endothelial surface inside the vascular lumen. The inhibitory effects of paCOS with F_A_ 0.46 on the intravasation of liver tumor cells were studied for the first time by using the tumor-vessel microsystem. The inhibitory effect of paCOS at 10 μg/mL on the intravasation of HepG2 cells was stronger than that of paCOS at 100 μg/mL (Figure 4), possibly due to the reason that paCOS had a destructive effect on tumor blood vessels at the concentration of 100 μg/mL, which was similar to 5-Fu at the same concentration. Thus both diffusion of paCOS to reach the tumor cells and the inhibitory effect of paCOS on liver tumor cells were affected. This result indicated that the paCOS concentration had a great inhibitory influence on the invasion of liver tumor cells into blood vessels. Matrix metalloproteinase-9 (MMP-9) is a vital component in cancer invasion and metastasis [33]. Previous studies on anti-angiogenic activity of COS showed that COS could inhibit the expression of MMP-9 [19,34]. Here, paCOS might also inhibit the intravasation of liver tumor cells by interfering with the activity of MMP-9 The inhibitory effect of paCOS on the intravasation of HepG2 cells was stronger than that on invasion of HepG2 cells (Figure 4 and Appendix A). These results suggested that endothelial cells with “blood flow” may be critical for the efficacy of paCOS on cancer metastasis, and functional vasculature and interstitial fluid flow may have major influence on drug transport [13].

Followed by intravasation, adhesion of the cancer cell onto the endothelium is the first step of the extravasation [5,35]. The inhibitory effects of paCOS with F_A_0.46 on the adhesion of liver tumor cells were studied for the first time on the tumor-vessel microsystem. paCOS with F_A_ 0.46 reduced the number of liver tumor cells adhering onto the surface of HUVECs layer after 3 h of intervention, indicating that paCOS might significantly inhibit the colonization of liver tumor cells and the formation of new metastases in a short period of time.

## 4. Materials and Methods 

### 4.1. Preparation of paCOS with F_A_ 0.46 from Partially Acetylated Chitosan

Partially acetylated chitosans were prepared from chitin in our lab by following the published procedure, and F_A_ was determined by ^1^H NMR [36,37]. paCOS was prepared by enzymatic hydrolysis of partially acetylated chitosan using recombinant endo-chitosanase (CSN) from *Aspergillus fumigatus* overexpressed in *Pichia pastoris* [25]. The hydrolysis products were analyzed by MALDI-TOF-MS (Agilent Technologies, Santa Clara, CA, USA). 

### 4.2. Fabrication and Assembly of the Microfluidic Device

The tumor-vessel microfluidic device used in this paper was fabricated with soft lithography using photoresist [38]. Briefly, all microchannel layers were individually prepared by casting PDMS prepolymer (10:1 w/w ratio of PDMS to curing agent) on a microfabricated mold of the inverse channel design made of photoresist, and curing the polymer at 60 °C for 12 h. The fluid microchannel (1 mm wide × 8 mm long × 0.1 mm high) and cell culture chambers (5 mm diameter × 2 mm high) in the upper and lower layers had the same size, and the diameter of the peripheral holes for connecting the tube was 1.5 mm. Porous polycarbonate (PC) membranes (7 mm diameter × 30 μm thickness) with a 10-μm pore size were placed between PDMS plates for on-chip cell culture. After careful alignment along the vertical direction, the PDMS plates were superimposed with the top and bottom PMMA frames, and fastened with screws. 

### 4.3. Cell Culturing and Establishment of the Tumor-Vessel Microsystem

Liver cancer cells (HepG2, SMMC-7721 cells and MHCC97-L) from American Type Culture Collection (Manassas, VA, USA) were cultured in MEM medium containing 10% (w/v) of fetal bovine serum and 1% (w/v) of non-essential amino acids (Gibco, Waltham, MA, USA). Endothelial cells (EC) EAhy 926 from American Type Culture Collection were cultured in DMEM/F12 medium (Gibco, Waltham, MA, USA) containing 10% (w/v) of fetal bovine serum. Penicillin (100 units/mL) and streptomycin (100 µg/mL) (Gibco, Waltham, MA, USA) were added to all aforementioned media. All cells were cultured in a cell incubator with 5% CO_2_ at 37 °C. 

After fabrication of the microdevice, the tubing and microfluidic channels were sterilized by high pressure steam (121 °C, 15 min) and drying the entire system in a 60 °C oven. After vascular endothelium (EAhy926) was digested from the plate, it was labelled with cell-tracker blue (Thermo Scientific, Waltham, MA, USA). Then, EAhy926 cells (1 × 10^6^ cells /mL) coated with collagen type I hydrogel (3 mg/mL) were seeded on the porous PC membrane and incubated at 37 °C for 3 h, allowing seeded endothelial cells to attach to the membrane surface. Then, the microdevice was assembled as described above. The tubes were connected to the inlet and outlet holes at the top of the PMMA frame. The culture medium was infused into the inlets from the medium tanks with a multi-channel peristaltic pump (WATSON MARLO, Wilmington, MA, USA) and flowed through both top and bottom layers in the microfluidic chip at a constant flow rate (120 µL/h), which produces 1.6 dyn/cm^2^ shear stress within the normal physiological range. After three days of cultivation, tumor cells (HepG2), which were stained by cell-tracker green (Thermo Scientific, Waltham, MA, USA) and wrapped by matrigel (BD Biosciences, New York, NY, USA), were seeded on the on the other side of the porous membrane and cultivated with endothelial cells on a microfluidic chip for 24 h, as shown in Appendix A.

### 4.4. MTT Assay

For static proliferation assay, MTT assay was done. Tumor cells were seeded in 96-well plates at a density of 5 × 10^3^ cells/well. After 24 h cultivation, cells were incubated in the culture medium with or without COS, paCOS (F_A_ = 0.46), chitin oligosaccharide (NACOS) and 5 fluorouracil (5-Fu) at the concentration of 100 μg/mL for another 24 h. Then, MTT (Sigma, St. Louis., MO, USA) solution (2 mg/mL) was added onto the plates, and incubated at 37 °C for 4 h. Formazan, derived from MTT by living cells, was dissolved in DMSO (150 μL per well), and the absorbance at 570 nm was measured. All MTT experiments were performed in octuplicate and repeated at least 3 times.

### 4.5. Tumor Cell Proliferation and Apoptosis Assay

For dynamic proliferation assay, liver cancer cells (HepG2) were co-cultured with endothelial monolayer at a density of 5 × 10^5^ cells/mL within the microsystem as described above. The culture medium in the absence or presence of various concentrations of paCOS (1, 10, 100 μg/mL) was flowed across two layers of endothelial cells providing nutrient and drugs for tumor cells. After the entire system was incubated at 37 °C for 24 h, the porous membrane was removed from the chip and the number of tumor cells in five different regions was counted under a fluorescence microscope.

For apoptosis assay, HepG2 cells were incubated in the culture medium with increasing concentrations of paCOS (1, 10, 100 μg/mL) for 24 h and then stained with Hoechst 33342 (Thermo Scientific, Waltham, MA, USA) at a concentration of 5 μg/mL for 30 min at 37 °C. Total numbers of cells in a field of microscope view (200×) were counted, and the percentage of apoptotic cells with condensed nuclei was quantified. 

### 4.6. Tumor Cell Migration and Wound-Healing Assay

For dynamic migration assay, six tumor cell clusters packaged by 8 mg/mL of matrigel (Thermo Scientific, Waltham, MA, USA) with a diameter of 0.5 mm were seeded on the endothelial monolayer within the microsystem. The culture medium in the absence or presence of various concentrations of paCOS (1, 10, 100 μg/mL) was flowed across two layers of endothelial cells providing nutrient and drugs for tumor cells. After the entire system was incubated at 37 °C for 18 h, the porous membrane was removed from the microfluidic chip and the number and the migration distance of tumor cells away from the original area was calculated under a fluorescence microscope.

For static migration assay, wound-healing assay was performed. Tumor cells stained by cell-tracker green at a density of 1 × 10^5^ cells/well were seeded in 24-well dishes and incubated for 24 h, monolayers were then disrupted with a cell scraper (1 mm wide). After 18 h incubation in the culture medium with or without various paCOS concentrations (1, 10, 100 μg/mL), photographs were taken under a fluorescence microscope to calculate the number and migration distance of tumor cells. Experiments were carried out in triplicate, and three fields were recorded for each well.

### 4.7. Tumor Cell Intravasation and Invasion Assay

For dynamic intravasation assay, liver cancer cells (HepG2) were co-cultured with endothelial monolayer at a density of 1 × 10^6^ cells/mL within the microsystem as described above. CXCL12 (Sigma, St. Louis., MO, USA) at a concentration of 100 ng/mL and various paCOS concentrations (1, 10, 100 μg/mL) were added to the culture medium flowed through the lumen of the blood vessel, and FITC-labeled 10 kDa dextrans (Sigma, St. Louis., MO, USA) at a concentration of 2 nmol/mL were perfused through the microchannel of the upper PDMS layer. After the entire system was incubated at 37 °C for 24 h, the porous membrane was removed from the chip and the number of tumor cells invading the vascular lumen was counted under a fluorescence microscope.

For static invasion assays, 5 × 10^4^ tumor cells stained by cell-tracker green (5 mM) were plated in the top chamber with matrigel coated membrane (24-well insert, pore size: 10 mm). Serum-free medium was added into the upper chamber, and the medium containing 10% FBS plus CXCL12 (100 ng/mL) was added into the lower chamber as a chemoattractant. After 24 h incubation in the culture medium with or without various paCOS concentrations (1, 10, 100 μg/mL) at 37 °C, the cells that invaded through the pores into the lower surface of the membrane were stained with haematoxylin and eosin and counted under a contrast inverted microscope. Three invasion chambers were used under each condition, and five fields were recorded for each well.

### 4.8. Tumor Cell Adhesion Assay

For dynamic adhesion assay, the culture medium in the absence or presence of various paCOS concentrations (1, 10, 100 μg/mL) was flowed across two endothelial cell layers, and the liver cancer cells (HepG2) at a density of 5 × 10^5^ cells/mL were added to above culture medium. After the entire system was incubated at 37 °C for 3 h, the porous membrane was removed from the chip and the number of tumor cells adhering to the surface of the underlying vascular endothelium was counted under a fluorescence microscope.

### 4.9. Paracellular Permeability Measurement

The barrier function of the endothelial monolayer formed by EAhy926 cells was evaluated by measuring the apparent permeability (*P_app_*) value of FITC-labeled dextrans (10 kDa) through the endothelial layer. After dynamic intravasation assay, one milliliter of D’Hanks solution (Thermo Scientific, Waltham, MA, USA) containing FITC-dextrans (10 kDa; 2 nmol/mL) was perfused through the microchannel of the upper PDMS layer, and the blank D’Hanks solution was circulated through the lower microchannel at a flow rate of 120 μL/h *P_app_* was calculated using the equation below [39]:
*P_app_(cm/s)* = (1/*AC*_0_)(*dQ/dt*)
where *A* = area of mass transfer, *C*_0_ = donor concentration of reagent in the upper medium, and *dQ/dt* = transmembrane transportation rate.

### 4.10. Haematoxylin and Eosin Staining Assay

The haematoxylin and eosin staining assay kit (Beyotimes, Xian, Shanxi, China) was used for observing the morphology of vascular endothelial cells (EAhy 926) and liver tumor cells (HepG2). After dynamic intravasation assay or static invasion assay, EAhy926 cells or HepG2 cells were washed with ice-cold phosphate buffered saline (PBS) and fixed with 10% paraformaldehyde. The cells were then stained with haematoxylin and eosin to improve visualization and were observed under a contrast inverted microscopy

### 4.11. Data Analysis and Quantification

All data were analyzed by averaging the values of at least three microfluidic devices, with each device representing one independent experiment. The mean value of each device was calculated based on the average of at least three parallel test. Student’s tests (three-sample, *p* < 0.05) and correlation analysis were performed with GraphPad software.

Tumor cells proliferation assay in the micro-device and MTT assay on the plate were done to evaluate the influence of the paCOS samples on the proliferation of HepG-2 cells. The proliferation of normal HepG-2 cells was set to 100%, as shown in Figure 3A and Appendix A.

## 5. Conclusions

In this study, a new paCOS with F_A_ 0.46 was prepared from partially deacetylated chitin with recombinant endo-chitosanase. With the use of a tumor-vessel microfluidic platform, it was observed for the first time that paCOS not only significantly inhibited the proliferation of liver tumor cells (HepG2) through vascular absorption, but also reduced the migration ability of liver tumor cells. Additionally, paCOS at 10 μg/mL had a stronger inhibitory effect on liver tumor cells invading blood vessels than at 100 μg/mL and paCOS at 100 μg/mL, which had a significant destructive effect on tumor vascular growth and barrier function. Besides, with only 3 h of treatment, paCOS significantly inhibited the adhesion of liver tumor cells onto the surface of HUVECs layer. Therefore, paCOS could potentially be used as anti-tumor metastasis reagents, and the tumor-vessel microfluidic platform could be used for investigating the mechanism of anti-metastasis drugs.

## Figures and Tables

**Figure 1 marinedrugs-17-00415-f001:**
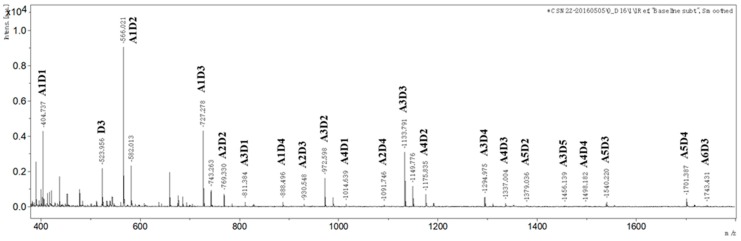
MALDI-TOF-MS analysis of paCOS (F_A_ = 0.46). *N*-acetyl glucosamine (GlcNAc) and glucosamine (GlcN) were represented by A and D respectively.

**Figure 2 marinedrugs-17-00415-f002:**
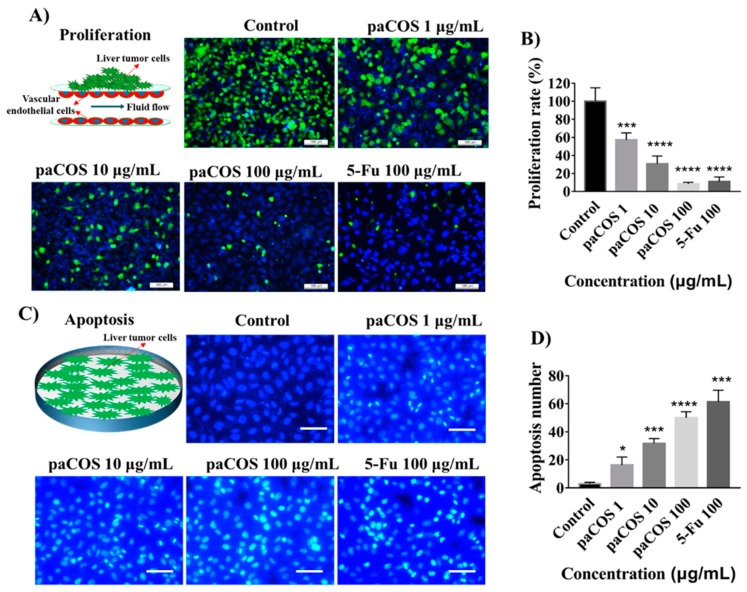
Inhibitory effects of paCOS with F_A_ 0.46 on liver tumor cell proliferation. Fluorescence views (scale bar: 100 μm) (**A**) and statistical analysis (**B**) of the proliferation rate of HepG2 cells (green) treated by paCOS with F_A_ 0.46 at different concentrations dissolved in culture medium on the tumor-vessel microsystem for 24 h. 5-Fu (100 μg/mL) was used as a positive control. Fluorescence views (scale bar: 100 μm) (**C**) and statistical analysis (**D**) of the number of HepG2 cells (blue) apoptosis treated by paCOS with F_A_ 0.46 at different concentrations dissolved in culture medium on the 96-well plates for 24 h. 5-Fu (100 μg/mL) was used as a positive control. Data are represented as the means ± SD (*n* = 5), * *p* < 0.05, *** *p* < 0.001, **** *p* < 0.0001.

**Figure 3 marinedrugs-17-00415-f003:**
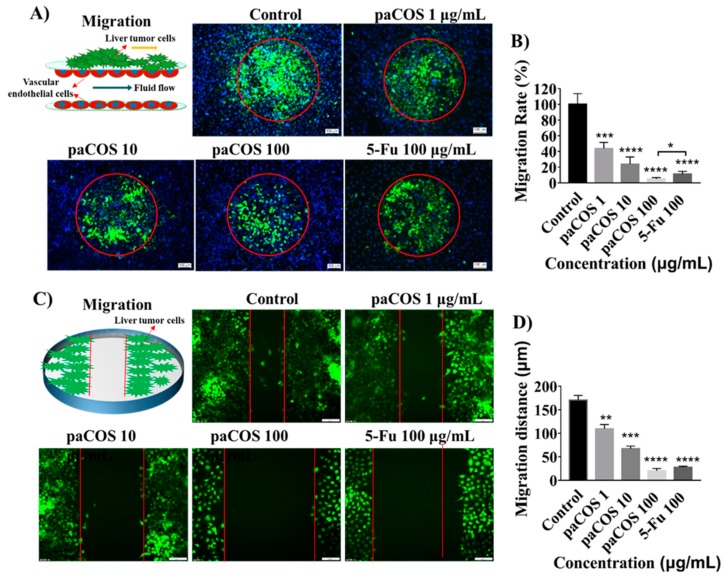
Inhibitory effects of paCOS with F_A_ 0.46 on liver tumor cell migration. Fluorescence views (scale bar: 100 μm) (**A**) and statistical analysis (**B**) of the migration rate of HepG2 cells (green) treated by paCOS (dissolved in culture medium) with F_A_ 0.46 at different concentrations on the tumor-vessel microsystem for 18 h. 5-Fu (100 μg/mL) was used as a positive control. Fluorescence views (scale bar: 100 μm) (**C**) and statistical analysis (**D**) of the migration distance of HepG2 cells (green) treated by paCOS (dissolved in culture medium) with F_A_ 0.46 at different concentrations on the 24-well plates for 18 h. 5-Fu (100 μg/mL) was used as a positive control. Data are represented as the means ± SD (*n* = 5), * *p* < 0.05, *** *p* < 0.001, **** *p* < 0.0001.

**Figure 4 marinedrugs-17-00415-f004:**
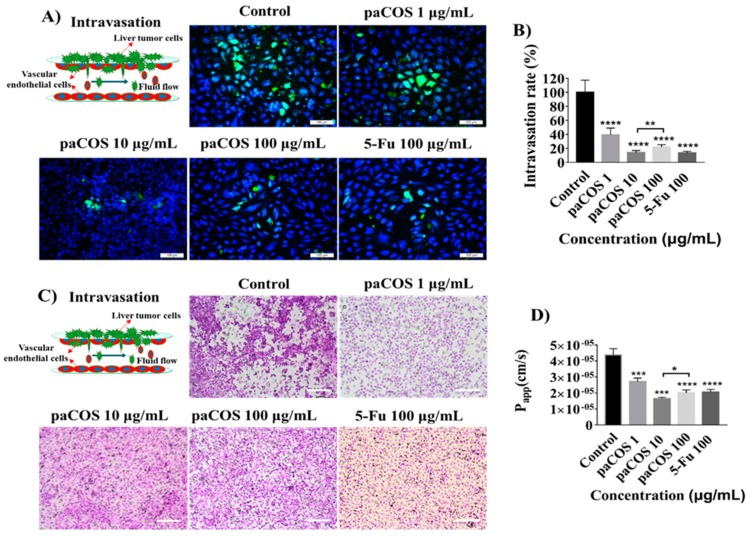
Inhibition effects of paCOS with F_A_ 0.46 on intravasation of liver tumor cells. Fluorescence views (scale bar: 100 μm) (**A**) and statistical analysis (**B**) of the intravasation rate of HepG2 cells (green) treated by paCOS with F_A_ 0.46 at different concentrations dissolved in culture medium on the tumor-vessel microsystem for 24 h. 5-Fu (100 μg/mL) was used as a positive control. (**C**) HE staining shows destruction of vascular endothelial cells (HUVECs) by liver tumor cells (HepG2) treated by paCOS with F_A_ 0.46 at different concentrations dissolved in culture medium on the tumor-vessel microsystem for 24 h (scale bar: 200 μm). (**D**) Statistical analysis of the *Papp* value of vascular endothelial layer (EAhy926) invaded by liver tumor cells (HepG2) treated by paCOS with F_A_ 0.46 at different concentrations dissolved in culture medium on the tumor-vessel microsystem for 24 h. 5-Fu (100 μg/mL) was used as a positive control. Data are represented as the means ± SD (*n* = 5), **P* < 0.05, *** *p* < 0.001, **** *p* < 0.0001.

**Figure 5 marinedrugs-17-00415-f005:**
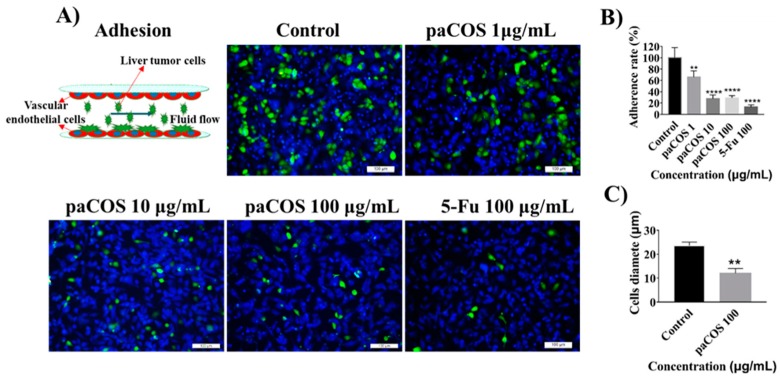
Inhibition effects of paCOS with F_A_ 0.46 on adhesion of liver tumor cells. Fluorescence views (scale bar: 100 μm) (**A**) and statistical analysis (**B**) of the adhesion rate of HepG2 cells (green) treated by paCOS with F_A_ 0.46 at different concentrations dissolved in culture medium on the tumor-vessel microsystem for 3 h. 5-Fu (100 μg/mL) was used as a positive control. (**C**) Statistical analysis of the diameter of HepG2 cells treated with or without paCOS at 100 μg/mL dissolved in culture medium on the tumor-vessel microsystem for 3 h. Data are represented as the means ± SD (*n* = 5), ** *p* < 0.01, **** *p* < 0.0001.

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
