# Peer review of "Inhibition of Liver Tumor Cell Metastasis by Partially Acetylated Chitosan Oligosaccharide on A Tumor-Vessel Microsystem"

_marinedrugs, 2019, doi:10.3390/md17070415_

Reviewer 1 Report

The manuscript titled: Inhibition of live tumor cell metastasis by partially acetylated chitosan oligosaccharide on tumor-vessel microsystem. the authors describe the synthesis and functional characterization of the therapeutic effect of paCOS in a microfluidic system using liver cells and endothelial cells.

The experimental design is sound but more cancer cells need to be included.

The manuscript needs extensive English language editing specifically the titles of the paragraphs.

The methods of the assays are not adequately described in the material and methods or the figure legends.

Reviewer 2 Report

This work analyses the inhibitory effect of paCOS on different stages of tumor metastasis using an in hose microdevise for a dynamic tumor-vessel microsystem. The work is well described, but control experiments to evaluate the effect of paCOS on the endothelial layer in a non-cancer cells model seems required to know if diffusion of paCOS through the endothelial barrier affects the properties of the endothelial cells.

- In the different cellular experiments (proliferation, migration, intravasation, adhesion), the buffer in which paCOS are dissolved is not indicated. Are they dissolved in buffer or culture medium?. Please indicate both in the figure legends and in the Materials and methods section.

- In the dynamic proliferation assay, are the endothelial cells affected by paCOS?. If they are affected, the diffusion of paCOS to reach the tumor cells may also be affected.

- The proliferation assay evaluates the effect of paCOS at different concentrations on the proliferation of liver tumor cells. A control with non-tumor cells (hepatocytes) is not indicated, which will inform about selectivity. The same controls with non-tumor cells will be needed for the apoptosis experiment.

- Effect of paCOS on the barrier function of vascular endothelial layer (line 161-65). Describe the HE staining (not defined). When determining the apparent permeability (Papp values), how the transmembrane transport rate was measured? (experimental section line 307).

-In the intravasation assay, a paCOS concentration of 10 uM  had the larger effect (Figure 4D), but the effect of paCOS on invasion  showed a different trend (Figure S3B). Please discuss.

- Line 175. It refers to Figure 5, which is missing. 

 It is not clear whether paCOS have an effect on the endothelial cells (toxicity, cell viability, permeability,....). Currently, the observed effects are assigned to the effect of paCOS to the tumor cells but it may also have a significant effect on the endothelial barrier. Additional control experiments are required.

Author Response

Round  2

Reviewer 1 Report

The authors responded to most of the raised concerns in the previous version. However, there are still some concerns:

The manuscript still needs substantial English languish editing.

The authors used 2 cell lines breast cancer cells (MDA-MB-231cells, ATCC) and colon cancer cells (Caco2 cells, ATCC), that are not liver cancer cell lines. The authors should revise the title of the manuscript to reflect the new findings.

Author Response

Dear reviewer:

On behalf of all contributing authors, I would like to express our sincere appreciation of your advice and constructive comments concerning our manuscript. These comments are valuable and helpful for improvement of our manuscript. According to your comments, we have revised the manuscript carefully. Our responses to your comments were listed as below. If there are any other comments, please let us know.

 Response to Reviewer 1 Comments

Point 1: The manuscript still needs substantial English languish editing.

Response: We appreciate the reviewer’s comments on these. As suggested by the reviewer, we have made substantial English languish editing in the revised manuscript.

Point 2: The authors used 2 cell lines breast cancer cells (MDA-MB-231cells, ATCC) and colon cancer cells (Caco2 cells, ATCC), that are not liver cancer cell lines. The authors should revise the title of the manuscript to reflect the new findings.

Response: We totally agree with the reviewer’s comment on this. As suggested by the reviewer, we have studied the inhibitory effects of paCOS at different concentration on the proliferation of two other liver cancer cells (SMMC-7721 cells from ATCC, MHCC97-L cells from ATCC) using the dynamic tumor-vessel microfluidic model. As shown in Figure S3 and 2B, it was found that the inhibitory effects of paCOS on SMMC-7721 cells and MHCC97-L cells were positively related to the concentrations of paCOS, and the proliferation rate of SMMC-7721 cells and MHCC97-L cells treated with paCOS (100 mg/mL) were 45.1 + 7.3 % and 55.2 + 2.6 %, which were lower than that on HepG2 cells (91.6 + 1.7 %).

Reviewer 2 Report

The authors have answered the comments and corrected the manuscript.
